# Analyzing Flow Cytometry or Targeted Gene Expression Data Influences Clinical Discoveries—Profiling Blood Samples of Pancreatic Ductal Adenocarcinoma Patients

**DOI:** 10.3390/cancers15174349

**Published:** 2023-08-31

**Authors:** Willem de Koning, Casper W. F. van Eijck, Fleur van der Sijde, Gaby J. Strijk, Astrid A. M. Oostvogels, Reno Debets, Casper H. J. van Eijck, Dana A. M. Mustafa

**Affiliations:** 1The Tumor Immuno-Pathology Laboratory, Department of Pathology & Clinical Bioinformatics, Erasmus University Medical Centre, 3015 GD Rotterdam, The Netherlands; w.dekoning.1@erasmusmc.nl (W.d.K.);; 2Clinical Bioinformatics Unit, Department of Pathology & Clinical Bioinformatics, Erasmus University Medical Centre, 3015 GD Rotterdam, The Netherlands; 3Department of Surgery, Erasmus University Medical Centre, 3015 GD Rotterdam, The Netherlands; 4Laboratory of Tumor Immunology, Department of Medical Oncology, Erasmus University Medical Centre, 3015 GD Rotterdam, The Netherlands

**Keywords:** pancreatic ductal adenocarcinoma (PDAC), FOLFIRINOX, flow cytometry, targeted gene expression, peripheral immune cell profile

## Abstract

**Simple Summary:**

We investigated the immunological changes in the blood of pancreatic ductal adenocarcinoma patients treated with a single cycle of FOLFIRINOX chemotherapy combined with lipegfilgrastim. We compared the use of flow cytometry and targeted gene expression analysis to study these immunological changes in blood samples. Our findings showed that FFX-Lipeg treatment increased the number of neutrophils and monocytes. Interestingly, flow cytometry analysis revealed an increase in B and T cells after treatment, while targeted gene expression analysis indicated a decrease in the expression of B and T cell-specific genes. This suggests that different measurement techniques can influence observed immunological changes. Therefore, the careful selection of an appropriate technique is essential when studying treatment effects in PDAC patients.

**Abstract:**

Introduction: Monitoring the therapeutic response of pancreatic ductal adenocarcinoma (PDAC) patients is crucial to determine treatment strategies. Several studies have examined the effectiveness of FOLFIRINOX as a first-line treatment in patients with locally advanced pancreatic cancer, but little attention has been paid to the immunologic alterations in peripheral blood caused by this chemotherapy regimen. Furthermore, the influence of the measurement type (e.g., flow cytometry and targeted gene expression) on the clinical discoveries is unknown. Therefore, we aimed to scrutinize the influence of using flow cytometry or targeted immune gene expression to study the immunological changes in blood samples of PDAC patients who were treated with a single-cycle FOLFIRINOX combined with lipegfilgrastim (FFX-Lipeg). Material and Methods: Whole-blood samples from 44 PDAC patients were collected at two time points: before the first FOLFIRINOX cycle and 14 days after the first cycle. EDTA blood tubes were used for multiplex flow cytometry analyses to quantify 18 immune cell populations and for complete blood count tests as the standard clinical routine. The flow cytometry data were analyzed with FlowJo software. In addition, Tempus blood tubes were used to isolate RNA and measure 1230 immune-related genes using NanoString Technology^®^. Data quality control, normalization, and analysis were performed using nSolver™ software and the Advanced Analysis module. Results: FFX-Lipeg treatment increased the number of neutrophils and monocytes, as shown by flow cytometry and complete blood count in concordance with elevated gene expression measured via targeted gene expression profiling analysis. Interestingly, flow cytometry analysis showed an increase in the number of B and T cells after treatment, while targeted gene expression analysis showed a decrease in B and T cell-specific gene expression. Conclusions: Targeted gene expression complements flow cytometry analysis to provide a comprehensive understanding of the effects of FFX-Lipeg. Flow cytometry and targeted gene expression showed increases in neutrophils and monocytes after FFX-Lipeg. The number of lymphocytes is increased after treatment; nevertheless, their cell-specific gene expression levels are downregulated. This highlights that different techniques influence clinical discoveries. Therefore, it is important to carefully select the measurement technique used to study the effect of a treatment.

## 1. Introduction

Pancreatic ductal adenocarcinoma (PDAC) is a malignancy that develops from the epithelial cells that line pancreatic ducts and is one of the most lethal cancer types [1]. The complex tumor (immune) microenvironment, the environment that surrounds the tumor cells, is composed of many cell types and the extracellular matrix. It includes immune cells, blood vessels, cancer-associated fibroblasts, and other cells that constantly interact with and influence each other. The combination of gemcitabine with nab-paclitaxel or a combination or the combined chemotherapeutic regimen of 5-fluorouracil, folinic acid, irinotecan, and oxaliplatin (FOLFIRINOX) is considered the first-line treatment option for locally advanced and metastatic PDAC [2]. For (borderline) resectable PDAC patients with a good performance status, FOLFIRINOX is the preferred adjuvant treatment and several randomized control clinical trials are currently investigating its applicability in the neoadjuvant setting [3]. However, FOLFIRINOX is often associated with toxicity, particularly neutropenia [3,4,5,6,7,8]. To prevent FOLFIRINOX-induced neutropenia, many clinicians consider the prophylactic administration of a granulocyte colony-stimulating factor (G-CSF), such as lipegfilgrastim, as standard therapy [9,10,11].

FOLFIRINOX has been shown to alter the intra-tumoral immune cell profile of PDAC patients. Increased effector T cells and reduced suppressor cells were reported in the pancreatic tumor after neoadjuvant FOLFIRINOX treatment [12]. Furthermore, FOLFIRINOX enhanced tumor antigen presentation, potentially synthesizing the pancreatic tumor for treatment with immune checkpoint inhibitors [13,14]. Nevertheless, the effect of FOLFIRINOX-lipegfilgrastim (FFX-Lipeg) on a patient’s peripheral immune profile remains unclear. More knowledge on the immunological changes caused by FOLFIRINOX could pave the way to improved immunotherapeutic approaches and is therefore of clinical interest.

Flow cytometry is a technique that is widely employed for evaluating the immune cell composition of peripheral blood. It utilizes laser-based technology to detect and analyze the chemical and physical properties of cells or particles present in fluid samples [15]. By leveraging flow cytometry, researchers can accurately quantify different immune cell populations within peripheral blood. This technique offers the advantage of providing a more comprehensive and detailed analysis at the single-cell level. Subpopulations can be distinguished although common protein markers are present, and cells can be assigned to subpopulations based on negative protein markers. However, flow cytometry does not allow for the robust quantification of subpopulations based on intracellular characteristics [16]. Furthermore, flow cytometry requires viable cells, which necessitates costly and time-consuming processing as well as storage procedures. For longitudinal sample collection, cryopreservation is often employed, but it has been observed to affect the expression of crucial markers for immune subsets [17,18,19].

RNA-based transcriptome analysis is an emerging technique that enables the study of diverse cellular processes, including immune responses and the identification of various cell types within the peripheral blood [20]. Targeted gene expression profiling provides a more comprehensive immune-related dataset compared to flow cytometry. The composition of the immune cell subpopulations with RNA-based transcriptome analysis is inferred from the generated bulk gene expression dataset and can be defined based on intracellular characteristics [21,22]. Furthermore, gene expression analysis is possible even when far fewer viable cells are available, as it only requires 25–100 ng of total RNA.

Recently, we identified the FOLFIRINOX delta Gene Expression Profiling (FFX-ΔGEP) score, which predicts the lack of a FOLFIRINOX response in PDAC patients after only one cycle of FFX-Lipeg treatment [23]. In this study, we aimed to scrutinize the influence of using flow cytometry or targeted immune gene expression to study the immunological changes in blood samples of PDAC patients who were treated with a single cycle of FFX-Lipeg. 

## 2. Material and Methods

### 2.1. Patient Cohort and Blood Collection

The 44 PDAC patients included in this study were hospitalized at the Erasmus University Medical Centre Rotterdam between February 2018 and October 2020. Fourteen of these patients had (borderline) resectable PDAC and participated in the PREOPANC-2 randomized clinical trial (Dutch trial register NL7094). Thirty patients participated in the iKnowIT prospective cohort study (Dutch trial register NL7522), of which 19 had locally advanced and 11 had metastasized PDAC. Exclusion criteria were <18 years of age, previous treatment with FOLFIRINOX, or co-treatment with another chemotherapeutic. The medical ethics committee of the Erasmus University Medical Centre Rotterdam approved both studies (MEC-2018-087 and MEC-2018-004), and patient samples were only used when written informed consent was provided. After the histological confirmation of the primary tumor or metastases, patients underwent treatment with FOLFIRINOX chemotherapy. Moreover, all patients received prophylactic treatment with the long-acting G-CSF lipegfilgrastim (Lonquex®; Teva Pharmaceuticals, Petach Tikva, Israel), administered 24 h after each cycle. This approach aimed to mitigate FOLFIRINOX-induced neutropenia and its associated complications [9,24]. Two types of whole-blood samples (2 EDTA and 1 Tempus) were collected from the 44 PDAC patients at two time points: at the baseline (on the same day, but before the first cycle) and 14 days after the first cycle, but before the second FOLFIRINOX cycle (Figure 1).

### 2.2. Flow Cytometry and Complete Blood Count

EDTA tubes were utilized to collect whole-blood samples for conducting flow cytometry and complete blood count (CBC) analyses. As part of the standard clinical routine, CBC tests were performed to evaluate lymphocyte, neutrophil, and thrombocyte counts. The whole-blood samples were subjected to multiplex flow cytometry to quantify 18 immune cell populations, following a previously established protocol [25,26]. Briefly, distinct immune cell subsets were individually identified and gated on a scatter plot based on CD45+ staining versus side scatter. Subsequently, specific markers further defined eosinophils (CD15+ CD16−), mature neutrophils (CD15high CD16high), immature neutrophils (CD15+ CD16+), classical monocytes (CD14+ CD16−), intermediate monocytes (CD14+ CD16+), non-classical monocytes (CD14− CD16+), dendritic cells (CD14− CD16− CD11c+), myeloid-derived suppressor cells (CD14+ CD16− CD11b+ HLA-DRlow), B cells (CD3− CD19+), natural killer cells (CD3− CD56+ CD16+/−), T cells (CD3+), γδ T cells (CD3+ TCRγδ+), CD4+ T cells (CD3+ TCRγδ− CD4+), and CD8+ T cells (CD3+ TCRγδ− CD8+). Absolute cell counts were determined using Flow-Count Fluorospheres (Beckman Coulter, Brea, CA, USA). FlowJo software (version 10.7, Tree Star, San Carlos, CA, USA) was used for data analysis.

### 2.3. Targeted Immune Gene Expression Profiling

The whole-blood samples intended for targeted gene expression analysis were collected in Tempus tubes (Applied Biosystems, Foster City, CA, USA) and subsequently stored at −80 °C. Tempus tubes are designed to contain an RNA-stabilizing reagent, which effectively preserves RNA quality, enabling the measurement of gene expression profiles without the need to isolate peripheral blood mononuclear cells [27]. To extract total RNA from the blood in Tempus tubes, a Tempus Spin RNA Isolation Kit from Thermo Fisher Scientific (Waltham, MA, USA) was employed, following the manufacturer’s instructions. Subsequently, RNA quality was assessed using an Agilent 2100 BioAnalyzer (Santa Clara, CA, USA). Samples with RNA concentrations below 35 mg/mL were excluded from further analysis. Corrected RNA concentrations were calculated based on the percentage of fragments within the 300–4000 nucleotide range to account for RNA degradation. An nCounter^®^ PanCancer Immune Profiling (IP) Panel and nCounter^®^ Myeloid Innate Immunity (MII) Panel, each consisting of 730 genes and 40 housekeeping genes, were used for NanoString targeted gene expression analysis of the 88 whole-blood samples. The IP panel targeted different immune-related pathways as well as immune and adaptive immune cell-related genes. The MII panel targeted genes involved in the innate immune response of myeloid-derived cells. For each sample, a total of 200 ng RNA, in a maximum volume of 7 μL, was subjected to hybridization with the two panels for a duration of 17 h at 65 °C, following the protocol provided by the manufacturer (NanoString Technologies Inc., Seattle, WA, USA). Subsequently, the nCounter^®^ FLEX platform was utilized to wash away the unbound probes and to count the genes by scanning 490 fields of view (FOV). Data quality control, normalization, and analysis were performed using nSolver™ software (version 4.0) and the Advanced Analysis module (version 2.0) of NanoString Technology Inc. [28]. Raw gene counts were normalized based on the most stable 19 housekeeping genes identified by the geNorm algorithm [29]. Subsequently, scaling was performed based on the 265 overlapping genes between the two panels, and all data were log_2_ transformed. Genes were included in the further analysis if their expression levels exceeded the limit of detection, which was calculated as the average count of the negative controls plus two standard deviations, in more than 50% of the gene expression profiles (4.35 log_2_). To identify differentially expressed genes (DEGs), simplified negative binomial models, a mixture of negative binomial models, or log-linear models were utilized, depending on the convergence of each gene. DEGs were defined as genes with a *p*-value < 0.05 after correction for multiple testing using the Benjamin–Hochberg (BH) method. To characterize the abundance of immune cell abundance, candidate cell-specific gene markers were identified as described previously [22]. The selection of marker genes was carried out by calculating pairwise similarity scores among the 61 detected candidate marker genes that exceeded the limit of detection. For each immune cell type, at least two unique marker genes with a pairwise similarity score above 0.6 were required. The abundance score of the immune cell types was calculated via the average marker expression. Pathway scores were extracted from the advanced analysis of nSolver software. Pathway enrichment analyses were performed with the differentially expressed genes (|Log_2_ fold change| > 0.5, P.BH < 0.05) using Metascape [30] and ClueGo [31].

### 2.4. Statistical Analysis

Statistical testing and data visualization were performed with R statistical software (v.4.1.2) [32]. We used paired two-sided Student *t*-tests as a parametric test. We used the R packages ggplot2 [33] and EnhancedVolcano [34] for data visualization. Heatmaps were generated using the Log_2_-normalized count data of significantly differentially expressed genes (|Log_2_ fold change| > 0.5, P.BH < 0.05). Genes that were determined to be outliers using Tukey’s rule were removed [35]. The heatmap was visualized using the web-based tool Morpheus by the Broad Institute (RRID: SCR_017386).

## 3. Results

### 3.1. Patients’ Characteristics

A total of 44 patients who received one FFX-Lipeg cycle were included. Three whole-blood samples (two EDTA, one Tempus) were collected at the baseline (on the same day before the first cycle) and 14 days after the first cycle but before the second cycle. The mean overall survival (95% CI) was 9 (11–14) months, calculated as the months between the first FFX-Lipeg cycle and the date of death. All clinicopathological characteristics are summarized in Appendix A.

### 3.2. FFX-Lipeg Therapy Results in Enhanced Frequencies of Granulocytes and Monocytes in the Blood

Flow cytometry analyses showed that one cycle of FFX-Lipeg significantly increased 17 out of 18 immune cell types. The most pronounced increase was observed in the number of granulocytes and monocytes (Figure 2). The only cell type that was not significantly altered was the CD16^+^ NK cells (Appendix A). Furthermore, the changes relative to the total leukocytes have been calculated (Appendix A). The CBC measurements showed a significant increase in lymphocytes and neutrophils but a significant decrease in thrombocytes after treatment (Appendix A).

The RNA samples for targeted immune expression profiling were of a minimum concentration of 35 mg/mL, meeting the requirement for targeted gene expression analysis. As a result, all of the samples included in the study exceeded the specified inclusion threshold. Targeted immune expression profiling showed that the total infiltration of CD45^+^ cells significantly increased after FFX-Lipeg treatment. The definition of infiltrated immune cells was based on 41 genes that showed a pairwise similarity higher than 0.6 and defined thirteen different immune cell types (Appendix A). The peripheral abundance of neutrophils and monocytes significantly increased after treatment. In contrast to the flow cytometry measurements, the peripheral abundance of T cells and B cells significantly decreased after treatment (Figure 3). The abundance of plasma B cells and mast cells was not significantly altered after treatment (Appendix A). Furthermore, the changes relative to the total leukocytes have been calculated (Appendix A).

### 3.3. FFX-Lipeg Results in an Increased Number of Lymphocytes, Yet Pathways Associated with Lymphocyte Functions Are Downregulated

Flow cytometry results showed that T, B, and NK cells, as well as their subtypes, were increased after FFX-Lipeg treatment. On the contrary, pathway analysis using nSolver software revealed decreased scores of natural killer, B cell, and T cell functions (Appendix A). On the other hand, data from both flow cytometry and targeted gene expression revealed an increase in the myeloid compartment after FFX-Lipeg therapy. In concordance with this, pathway analysis resulted in an increased score of the differentiation and maintenance of myeloid cells after FFX-Lipeg treatment (Figure 4).

Metascape enrichment analysis using the 39 downregulated (P.BH ≤ 0.05 and log2 fold of change ≤ −0.5) genes showed that the FFX-Lipeg treatment negatively affected the MHC class II antigen presentation (Appendix A). The 170 upregulated (P.BH ≤ 0.05 and log_2_ fold of change ≥ 0.5) genes showed an enriched neutrophil degranulation, IRAK4 deficiency (TLR5), leukotriene metabolic pathway, IL-4 signaling pathway, and activation of matrix metalloproteinases (Appendix A). ClueGo functional analysis using the 39 downregulated (P.BH ≤ 0.05 and log2 fold of change ≤ −0.5) showed a negative regulation of T cell-mediated immunity and an innate immune response (Appendix A). On the contrary, myeloid and neutrophil cell-related functions were stimulated, and genes related to lymphocyte proliferation were enriched in the 170 upregulated (P.BH ≤ 0.05 and log_2_ fold of change ≥ 0.5) genes (Appendix A).

In order to scrutinize the differences in the results between flow cytometry and gene expression, we performed a correlation analysis of proteins targeted by flow cytometry with their corresponding marker genes in the targeted immune gene expression profile for the two time points separately. Lymphocyte subtypes showed a higher correlation before FFX-Lipeg treatment in comparison to the correlation after treatment (Appendix A).

### 3.4. FFX-Lipeg Induces a Distinct Gene Expression Profile in the Peripheral Blood

A single FFX-Lipeg cycle induced profound genetic alterations in (Figure 5A). Among the 870 genes that surpassed the detection limit, 209 genes exhibited differential expression (|Log_2_ fold change| > 0.5, P.BH < 0.05). Specifically, following FFX-Lipeg treatment, 170 genes were found to be upregulated and 39 genes were downregulated (Figure 5B).

The most upregulated differentially expressed genes include matrix metalloproteinase-8 (MMP8, P.BH = 3.38 × 10^−20^, FOC = 4.99), lactotransferrin (LTF, P.BH = 1.09 × 10^−18^, FOC = 3.97) and carcinoembryonic antigen-related cell adhesion molecule 8 (CEACAM8, P.BH = 1.58 × 10^−19^, FOC = 3.16), which are mainly expressed by neutrophils. MMP8 plays a role in the degradation of the extracellular matrix by cleaving various substrates, including collagens and cytokines [36,37,38]. LTF is an iron-binding protein elevated specifically in polymorphonuclear neutrophils [39]. CEACAM8, also referred to as CD66b, is a surface glycoprotein that participates in heterophilic cell adhesion within activated neutrophils, along with other carcinoembryonic antigen-related cell adhesion molecules such as CEACAM6 [40]. Furthermore, integrin subunit beta 4 (ITGB4, P.BH = 8.05 × 10^−20^, FOC = 3.29) and lipocalin 2 (LCN2, P.BH = 3.12 × 10^−18^, FOC = 3.35) were also found to be upregulated. ITGB4 promotes cell migration and invasion in pancreatic cancer; however, its exact role in these processes remains unclear [41]. The iron-trafficking protein LCN2 plays a role in various biological processes, including apoptosis, innate immunity, and renal development [42]. 

On the other hand, (cytotoxic) T cell- and natural killer cell-specific genes, RUNX family transcription factor 3 (RUNX3, P.BH = 1.37 × 10^−12^, FOC = −0.549), granzyme B (GZMB, P.BH = 2.43 × 10^−11^, FOC = −0.527), and killer cell lectin like receptor D1 (KLRD1, P.BH = 1.96 × 10^−11^, FOC = −0.525) were downregulated consistently in all samples after treatment. Furthermore, HLA class genes, like major histocompatibility complex, class II, DP beta 1 (HLA-DPB1, P.BH = 1.30 × 10^−11^, FOC = −0.704), and major histocompatibility complex, class II, DR beta 3 (HLA-DRB3, P.BH = 1.96 × 10^−11^, FOC = −0.704) were also downregulated.

## 4. Discussion

In this study, we used paired blood samples of 44 FFX-Lipeg-treated PDAC patients to scrutinize the influence of using flow cytometry or targeted immune gene expression to study immunological changes. Flow cytometry and targeted gene expression profiling revealed a similar effect caused by a single cycle of FFX-Lipeg regarding granulocytes and monocytes; however, the measurement technique affects the observed changes regarding lymphocytes.

FFX-Lipeg treatment increased the total number of leukocytes, as shown by flow cytometry in concordance with elevated gene expression measured via targeted gene expression. As parts of these leukocytes, the number of neutrophils and monocytes as well as their cell-specific gene expression are increased; this is also shown through CBC. Contrarily, previously reported results showed a reduction in granulocytes and monocytes due to FOLFIRINOX treatment [12]. Nevertheless, studies on the effect of modified FFX-Lipeg show an increase in granulocytes, caused by lipegfilgrastim. The addition of lipegfilgrastim has been shown to decrease the incidence of neutropenic events and prolong the progression-free survival of patients [43,44]. 

Interestingly, flow cytometry analysis showed an increase in the number of B and T cells after treatment, while targeted gene expression analysis showed a decrease in B and T cell-specific gene expression. Both granulocytes and monocytes are potent suppressors of T cell functions and inhibit antitumor immune responses [45,46]. This could explain that even though the number of lymphocytes showed an increase after treatment in the flow cytometry data and complete blood count measurements, there is a decrease in cell-specific gene expression based on the gene expression analysis.

It is essential to highlight that the predominant share of leukocytes comprises granulocytes, a phenomenon amplified through Lipeg treatment. Consequently, adjusting cell prevalence through total leukocyte count could obscure the impact on other cell subsets. This could elucidate the apparent proportional reduction in monocytes and lymphocytes. This discrepancy contradicts the targeted gene expression outcomes relating to monocytes in comparison to the entirety of CD45. The correlation analysis highlights that the effect of FFX-Lipeg therapy influences the number of cells differently than cell-specific gene expression. This could indicate that, regardless of the number of lymphocytes, the function of those immune cells cannot be fulfilled. The pathway analyses further highlighted this downregulation of the functions of the lymphocytes. Based on the gene expression analysis, it seems that lymphocytes have a lower expression of functional genes after FFX-Lipeg treatment. 

In this study, we measured 1230 immune-related genes via targeted gene expression profile and 18 immune cell types via flow cytometry. The added value of targeted gene expression analysis is shown by the discovery that a single FFX-Lipeg cycle changed the expression of 209 immune-related genes significantly and caused a distinct genetic profile between the samples before and after treatment. Targeted gene expression profiles that specifically measure part of the immune-related genes enabled us to identify an FFX-ΔGEP score to predict the lack of a treatment response after a single FFX-Lipeg cycle, as has been described previously [23]. As far as we know, measuring with flow cytometry did not lead to a similar discovery. 

Besides a biological explanation, several technical factors might influence the discrepancy between the two measurement techniques. The higher detection of lymphocytes measured via flow cytometry could be due to non-specific bindings of lower-quality antibodies, whereas the lower detection of immune cell-specific gene expression could be caused by the amount of RNA present in the samples, as well as mRNA stability. Furthermore, the comparison was performed on heterogenous samples and different types of blood samples (EDTA/Tempus).

## 5. Conclusions

Flow cytometry could be used for the precise quantification of immune cell populations, whereas gene expression analysis gave a broader understanding of the immune expression activity of those cells. This highlights that measuring the number of cells in the blood does not reflect the immune functionality of these cells. To study the effects of treatment, different techniques must be used to obtain a more complete overview. This study revealed that measurement techniques influence clinical discoveries. 

## Figures and Tables

**Figure 1 cancers-15-04349-f001:**
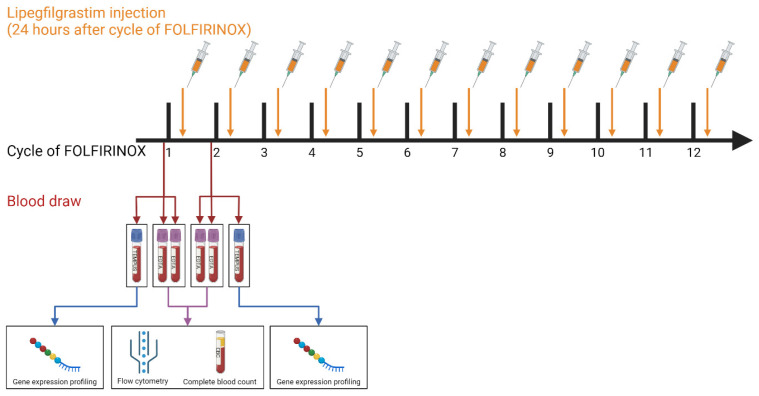
Schematic overview of the sample collection and measurements. The cycles of FOLFIRINOX chemotherapy (black), lipegfilgrastim injections (orange), and the blood draw time points (red). The blood is collected in EDTA and Tempus tubes. After the blood draw, the flow cytometry and complete blood count procedures are performed on the EDTA tubes. The Tempus tubes are used for gene expression profiling.

**Figure 2 cancers-15-04349-f002:**
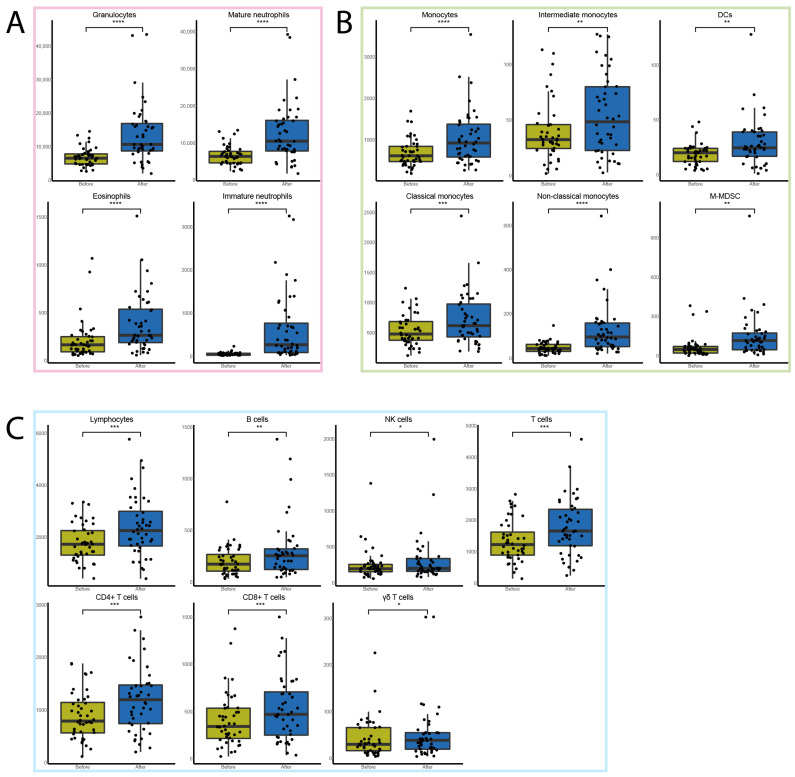
The effect of one cycle of FFX-Lipeg on the immune cells measured via flow cytometry. (**A**) The number of granulocytes (pink) including the subtypes that were significantly increased after treatment (blue) compared with before treatment (yellow). (**B**) The number of monocytes (green) including the subtypes that were significantly increased after treatment. (**C**) The number of lymphocytes (light blue) including the subtypes that were significantly increased after treatment. * *p* ≤ 0.05, ** *p* ≤ 0.01, *** *p* ≤ 0.001, **** *p* ≤ 0.0001.

**Figure 3 cancers-15-04349-f003:**
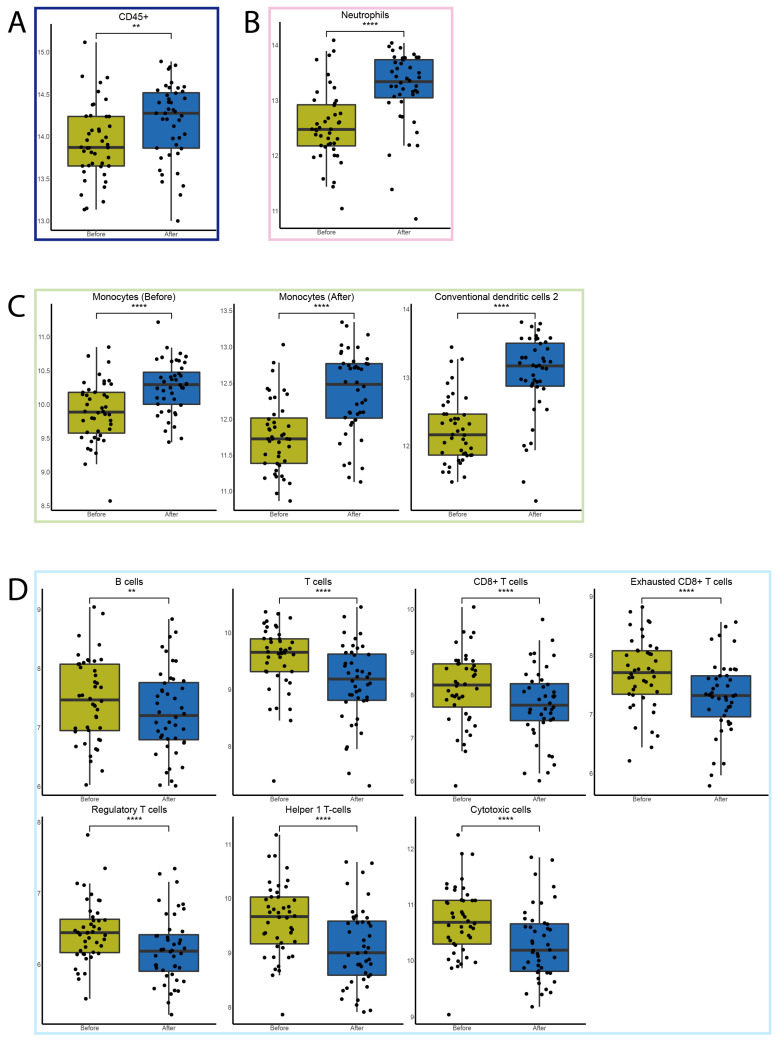
The effect of one FFX-Lipeg cycle on the immune cells measured by targeted gene expression. (**A**) The total immune cell (dark blue) abundance was significantly increased after treatment (blue) compared with before treatment (yellow). (**B**) The abundance of granulocytes’ (pink) subtype neutrophils was significantly increased after treatment. (**C**) The abundance of monocytes (green) including the subtypes was significantly increased after treatment. (**D**) The abundance of lymphocytes (light blue) including the subtypes was significantly decreased after treatment. ** *p* ≤ 0.01, **** *p* ≤ 0.0001.

**Figure 4 cancers-15-04349-f004:**
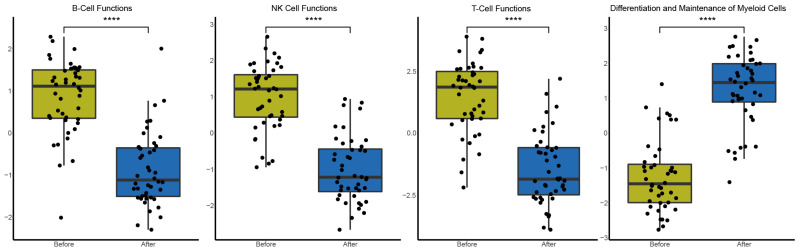
Differential expression of predefined pathway genes after one cycle of FFX-Lipeg. A significant upregulation of leukocyte functions and the differentiation as well as maintenance of myeloid cells was measured. On the contrary, natural killer, B cell, and T cell functions were significantly downregulated. **** *p* ≤ 0.0001.

**Figure 5 cancers-15-04349-f005:**
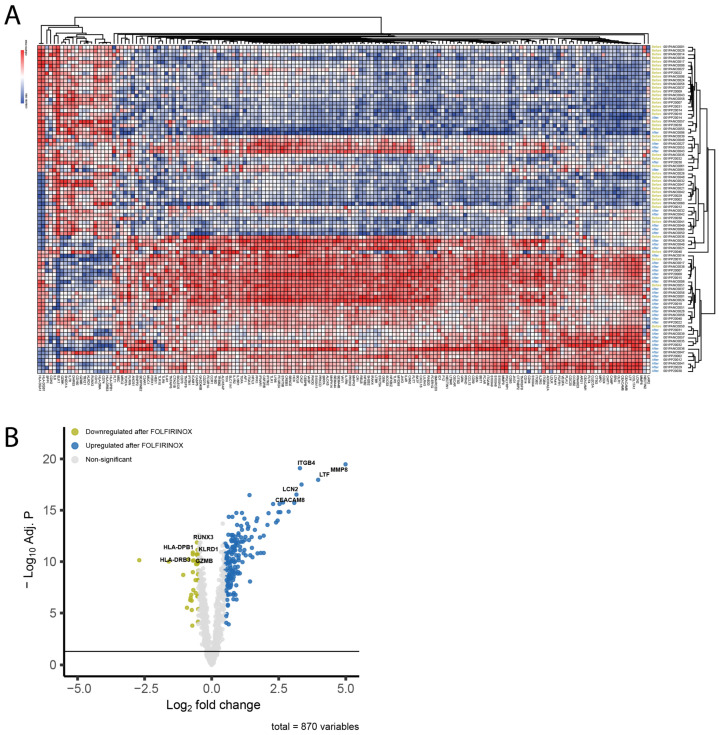
The effect of one FFX-Lipeg cycle on the gene expression profile. (**A**) The clustering of the genes showed an almost distinct genetic profile after one cycle. (**B**) A volcano plot highlighting the genes that were significantly altered (243 genes upregulated and 213 genes downregulated) after one cycle.

## Data Availability

The NanoString data presented in the study are deposited in the Gene Expression Omnibus (GEO) repository, accession number GSE241957.

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
