# Peer review of "Analyzing Flow Cytometry or Targeted Gene Expression Data Influences Clinical Discoveries—Profiling Blood Samples of Pancreatic Ductal Adenocarcinoma Patients"

_cancers, 2023, doi:10.3390/cancers15174349_

Round 1
Reviewer 1 Report
The authors applied two different technological approaches, namely flow cytometry and targeted gene expression, to evaluate the impact of FFX-Lipeg treatment on peripheral blood leukocytes from pancreatic ductal adenocarcinoma patients.
They studied 44 patients before and 14 days after treatment (before the second administration).
The authors analyzed 18 immune cell populations by flow and 1230 immune-related genes.
This is a well written manuscript but, despite the study and some results sounds interesting, I have some major concerns.
Major concerns:
1. The authors did not include an age- and gender-matched heathy control group that could allow the comparison of the results obtained in patients group before treatment with those from the healthy group.
2. The authors must present the results of the 18 populations measured by flow in frequency (%) among total leukocytes, and compare the results before and after treatment. The differences obtained before and after treatment in terms of cells per microliter, are due to the increase of leukocytes induced by lipegfilgrastim.
It is crucial to understand if the frequency of B cells and T cells decrease which will be on line with the decrease observed in gene expression.
3. The authors must inform how many samples with RNA concentrations below 35 mg/mL were excluded, to have a better interpretation of the results, and if it is possible to compare flow with targeted gene expression.
4. In my opinion, the more interesting results are those obtained in targeted gene expression, therefore the manuscript must be completely rewritten to put all the focus on the immune function modulation induced by the treatment, and if these results present any correlation with disease stage, treatment response and overall survival. Particularly, the results obtained in T cell functional compartments (Th1, Tregs, cytotoxic T cells, exhausted, etc), and in the different monocytes and dendritic cells subpopulations.
5. In figure 3 and 4, the authors must indicate the evaluated genes for each cell population (figure 3) or for leukocyte functions (figure 4).
6. In legend of figure 3, the authors wrote that the abundance of mast cells (yellow) was significantly increased after treatment, but it looks on contrary. Moreover, the results on mast cells are not in agreement to those presented in supplementary figures, in terms of significance. In the same legend (D) the authors said that “the abundance of monocytes (green) including the subtypes was significantly increased after treatment”, but we can´t follow this information in the graphs, since we only have the information about total monocytes and also from conventional dendritic cells 2, and which means before and after in parenthesis at the top of the graphics.
7. How the authors explain the results of gene expression obtained in mast cells, particularly before treatment, since mast cells are almost absent in peripheral blood?
8. Have the authors evaluated genes related to hematopoietic precursors, since filgrastim induces an increase of CD34+ cells in peripheral blood? If so, those results could be presented.
Author Response
Reviewer 1
We thank the reviewer for the feedback that helped to improve the manuscript. We have changed the manuscript with track changes according to the feedback and will describe this per comment.
The authors applied two different technological approaches, namely flow cytometry and targeted gene expression, to evaluate the impact of FFX-Lipeg treatment on peripheral blood leukocytes from pancreatic ductal adenocarcinoma patients.
They studied 44 patients before and 14 days after treatment (before the second administration).
The authors analyzed 18 immune cell populations by flow and 1230 immune-related genes.
This is a well written manuscript but, despite the study and some results sounds interesting, I have some major concerns.
Major concerns:
- The authors did not include an age- and gender-matched heathy control group that could allow the comparison of the results obtained in patients group before treatment with those from the healthy group.
We thank the reviewer for this comment. We have used paired samples, where the first sample was collected before treatment, establishing them as baseline measurements. This strategic approach enhances the statistical robustness of our tests. A healthy age- and gender-matched control group could not undergo testing, as administering FFX-Lipeg to them is not feasible.
- The authors must present the results of the 18 populations measured by flow in frequency (%) among total leukocytes, and compare the results before and after treatment. The differences obtained before and after treatment in terms of cells per microliter, are due to the increase of leukocytes induced by lipegfilgrastim.
It is crucial to understand if the frequency of B cells and T cells decrease which will be on line with the decrease observed in gene expression.
We have decided to perform the suggested analysis as we agree that the total number of leukocytes might influence the conclusions drawn from the data. Nevertheless, especially regarding B and T cells, the large number of Granulocytes and Monocytes present in the samples skew the lower number of B and T cells when correcting for the total number of cells.
- The authors must inform how many samples with RNA concentrations below 35 mg/mL were excluded, to have a better interpretation of the results, and if it is possible to compare flow with targeted gene expression.
Every sample included in our study surpassed the defined inclusion threshold. We thought it is important to add the RNA concentration criterium to share one of the limitations when measuring by targeted gene expression profile. However, we understand that can be a bit confusing. Therefore, we rephrased that sentence, and now reads as “RNA samples were of a minimum concentration of 35 mg/mL, meeting the requirement for targeted gene expression analysis. As a result, all samples included in the study exceeded the specified inclusion threshold.” Page 6, lines 223-225.
- In my opinion, the more interesting results are those obtained in targeted gene expression, therefore the manuscript must be completely rewritten to put all the focus on the immune function modulation induced by the treatment, and if these results present any correlation with disease stage, treatment response and overall survival. Particularly, the results obtained in T cell functional compartments (Th1, Tregs, cytotoxic T cells, exhausted, etc), and in the different monocytes and dendritic cells subpopulations.
We thank the reviewer for this point and share that targeted gene expression profiles are very important results. Nevertheless, it's essential to clarify that the primary objective of our research was to showcase the potential impact of the chosen measurement technique on the derived conclusions. Rather than directly comparing flow cytometry with targeted gene expression profiling, we intended to raise awareness among researchers and the medical community about the critical need for a deliberate selection of techniques. Our focus lies in encouraging meticulous technique selection.
It's worth noting that we've previously published an article centered around the immune function modulation prompted by the treatment. This article specifically explores any potential correlations between the immune-related findings and factors such as disease stage, treatment response, and overall survival, utilizing the approach of targeted gene expression profiling. https://doi.org/10.1016/j.ejca.2022.12.024.
- In figure 3 and 4, the authors must indicate the evaluated genes for each cell population (figure 3) or for leukocyte functions (figure 4).
We have added the requested genes for Figures 3 & 4 respectively as Supplementary Tables 2 & 3.
- In legend of figure 3, the authors wrote that the abundance of mast cells (yellow) was significantly increased after treatment, but it looks on contrary. Moreover, the results on mast cells are not in agreement to those presented in supplementary figures, in terms of significance. In the same legend (D) the authors said that “the abundance of monocytes (green) including the subtypes was significantly increased after treatment”, but we can´t follow this information in the graphs, since we only have the information about total monocytes and also from conventional dendritic cells 2, and which means before and after in parenthesis at the top of the graphics.
We would like to thank the reviewer for pointing out the contradictions in our manuscript. We have double-checked the numbers and have adjusted the legend and Supplementary Figure 3 accordingly. Regarding Figure 3D the parenthesis refers to different monocyte definitions based on pairwise similarity in the before and after treatment group. This should now be more apparent by providing the cell definitions in Supplementary Table 2.
- How the authors explain the results of gene expression obtained in mast cells, particularly before treatment, since mast cells are almost absent in peripheral blood?
Another very important question that we appreciate. We think that mast cells can be measured in the blood, although they are relatively rare in circulation compared to their presence in tissues. For example, they can be detected when performing a complete blood count with differential (CBC with differential), which provides information about various types of blood cells, including white & red blood cells, and platelets. For generating targeted gene expression in blood samples, we used a very sensitive technique that can detect very low levels of genes up to 0.125FM of RNA. Therefore, we think that marker genes of mast cells can be detected in blood samples using Nanostring technology.
- Have the authors evaluated genes related to hematopoietic precursors, since filgrastim induces an increase of CD34+ cells in peripheral blood? If so, those results could be presented.
We would like to thank the reviewer for this suggestion. Unfortunately, the gene expression marker for CD34 is only detected in one sample (before treatment). Therefore, statistical testing cannot be performed. Furthermore, similar markers were not included in the flow cytometry analysis. Therefore, analyzing hematopoietic cells does not add to achieving the aim of the study.

Reviewer 2 Report
The authors of the manuscript entitled " Analyzing flow cytometry or targeted gene expression data influences clinical discoveries; profiling blood samples of pancreatic ductal adenocarcinoma patients" have done a commendable job in showing the immunological changes in blood of pancreatic adenocarcinoma patients treated with Folfirinox chemotherapy combined with lipegfilgrastim. Additionally, they used flow cytometry and targeted gene expression analysis to study immunological changes in blood samples. The treatment with drug mentioned above led to increase in the number of neutrophils and monocytes. Interestingly their study shows a decrease in the expression of B and T cells specific genes but show an increase B and T cells by flow cytometry analysis. The authors also used blood samples from 44 FFX-Lipeg treated PDAC patients to study immunological changes. The authors also measured about 1230 immune related genes by targeted gene expression and 18 immune cell types by flow cytometry. Finally the authors conclude that flow cytometry can be used for precise quantification of immune cell populations whereas gene expression gave a broader understanding of the immune expression activity in those cells suggesting that measuring the number of cells in the blood does not have to reflect the immune functionality of the cells. This study directs the need for complementary measuring techniques of any therapeutic strategies that can accurately draw out the immunological changes observed especially in the treatment of PDAC patients. The manuscript is well written.
Author Response
Reviewer 2
The authors of the manuscript entitled " Analyzing flow cytometry or targeted gene expression data influences clinical discoveries; profiling blood samples of pancreatic ductal adenocarcinoma patients" have done a commendable job in showing the immunological changes in blood of pancreatic adenocarcinoma patients treated with Folfirinox chemotherapy combined with lipegfilgrastim. Additionally, they used flow cytometry and targeted gene expression analysis to study immunological changes in blood samples. The treatment with drug mentioned above led to increase in the number of neutrophils and monocytes. Interestingly their study shows a decrease in the expression of B and T cells specific genes but show an increase B and T cells by flow cytometry analysis. The authors also used blood samples from 44 FFX-Lipeg treated PDAC patients to study immunological changes. The authors also measured about 1230 immune related genes by targeted gene expression and 18 immune cell types by flow cytometry. Finally the authors conclude that flow cytometry can be used for precise quantification of immune cell populations whereas gene expression gave a broader understanding of the immune expression activity in those cells suggesting that measuring the number of cells in the blood does not have to reflect the immune functionality of the cells. This study directs the need for complementary measuring techniques of any therapeutic strategies that can accurately draw out the immunological changes observed especially in the treatment of PDAC patients. The manuscript is well written.
We thank the reviewer for the positive feedback on our manuscript and the accurate summary provided. We are very pleased with this opinion.

Reviewer 3 Report
Specific comments to the authors
The authors Willwm de Koning et al. of the submitted manuscript "Analyzing flow cytometry or targeted gene expression data influences clinical discoveries; profiling blood samples of pancreatic ductal adenocarcinoma patients" investigated possible immunological changes by flow cytometry and targeted gene expression in the blood of patients with PDAC treated with a single cycle of FOLFIRINOX chemotherapy combined with lipegfilgrast.
In summary, based on their investigations, the authors were able to show that (i) flow cytometry analysis indicated an increase in neutrophils and monocytes as well as B and T cells, (ii) whereas targeted gene expression analysis showed an increase in neutrophil and monocyte genes as well as a decrease in B and T cell-specific gene expression. Therefore, the authors postulate that the combination of flow cytometry analysis and targeted gene expression may provide comprehensive insights into the potential immunological treatment effects of FFX-Lipeg in patients with PDAC.
Overall, the manuscript provides some interesting information on immune-related effects in patients with PDAC treated with a single cycle of FOLFIRINOX chemotherapy combined with Lipegfilgrast. The manuscript (including the presentation) is clear and convincing. Methods are well described. Although the results and discussion are clearly presented, the authors (see specific comments) need to clarify definitively whether the authors are focusing more on technical or immunological aspects of the presented study, which is not definitively clear so far. In conclusion, the data presented contain some interesting data. After taking into account the above-mentioned specific comments (see below), the manuscript has the potential to be accepted.
Specific comments
Title: The title does not reflect all facets of the presented study.
Abstract: The conclusion does not really interpret or discuss the results.
Introduction: The term "ecosystem" is not appropriate in the context of a solid tumour.
Materials and methods: The manuscript contains "blind" references ("Error! Reference source not found") throughout the manuscript. Please correct these appropriately. The rationale for the flow cytometry markers used should be given. Why not markers for Th1, Th2 or Th17 differentiation in correlation with the targeted gene expression analysis for Th1?
Results:
# Figure 1: Why are the blood samples taken at these two times and not more often? Please explain.
# Figures 2, 3 and 4: The flow cytometry and targeted gene expression data should be statistically analysed using correlation techniques. Furthermore, the immunological findings should also be correlated with the in-situ analysis of the tumour specimen.
Discussion: With regard to the results, the authors should discuss the findings at the cellular and molecular level in more detail. Specifically, the increase in the immature system could be interpreted as a "normal" response to the cytotoxic effects of FOLFIRINOX chemotherapy combined with lipegfilgrast. On the other hand, the decrease at the mRNA level could be interpreted as an immunosuppressive function of the chemotherapy used. Finally, the authors should mention that the results must be related to deep in-situ characterizations of tumour specimens.
Minor editing of English language required.
Author Response
Reviewer 3
We thank the reviewer for the feedback that helped to improve the manuscript. We have changed the manuscript with track changes according to the feedback and will describe this per comment.
Specific comments to the authors
The authors Willwm de Koning et al. of the submitted manuscript "Analyzing flow cytometry or targeted gene expression data influences clinical discoveries; profiling blood samples of pancreatic ductal adenocarcinoma patients" investigated possible immunological changes by flow cytometry and targeted gene expression in the blood of patients with PDAC treated with a single cycle of FOLFIRINOX chemotherapy combined with lipegfilgrast.
In summary, based on their investigations, the authors were able to show that (i) flow cytometry analysis indicated an increase in neutrophils and monocytes as well as B and T cells, (ii) whereas targeted gene expression analysis showed an increase in neutrophil and monocyte genes as well as a decrease in B and T cell-specific gene expression. Therefore, the authors postulate that the combination of flow cytometry analysis and targeted gene expression may provide comprehensive insights into the potential immunological treatment effects of FFX-Lipeg in patients with PDAC.
Overall, the manuscript provides some interesting information on immune-related effects in patients with PDAC treated with a single cycle of FOLFIRINOX chemotherapy combined with Lipegfilgrast. The manuscript (including the presentation) is clear and convincing. Methods are well described. Although the results and discussion are clearly presented, the authors (see specific comments) need to clarify definitively whether the authors are focusing more on technical or immunological aspects of the presented study, which is not definitively clear so far. In conclusion, the data presented contain some interesting data. After taking into account the above-mentioned specific comments (see below), the manuscript has the potential to be accepted.
We appreciate the reviewer's insightful overview and perspective. By elaborating that the study's primary focus is on technical aspects rather than solely on immunological or biological factors, several questions and comments are likely to gain greater clarity.
Specific comments
Title: The title does not reflect all facets of the presented study.
We have thought about the title. We think it reflects the study, therefore, we didn’t change it. We hope that the reviewer agrees, especially after elaborating on the technical aspect of the study.
Abstract: The conclusion does not really interpret or discuss the results.
We have reevaluated our conclusions. We appreciate if the reviewer could reassess the conclusions from a technical aspect rather than the immunological aspect.
Introduction: The term "ecosystem" is not appropriate in the context of a solid tumour.
We have replaced the word “ecosystem” with “environment”.
Materials and methods: The manuscript contains "blind" references ("Error! Reference source not found") throughout the manuscript. Please correct these appropriately.
We have updated the references to the correct Figures.
The rationale for the flow cytometry markers used should be given. Why not markers for Th1, Th2 or Th17 differentiation in correlation with the targeted gene expression analysis for Th1?
We thank the reviewer for this question. We completely agree that Th1, Th2 and Th17 are very important cell types to be investigated. However, the flow cytometry panels were optimized for a big study. We utilized the information that was permitted to be shared with us, without affecting the other ongoing investigations. In addition, using the panCancer immune profiling targeted gene expression panel, we are not able to identify Th1, Th2 and Th17 accurately. Therefore, adding them to the flow cytometry results will not help achieve the aim of this work.
Results:
# Figure 1: Why are the blood samples taken at these two times and not more often? Please explain.
We thank the reviewer for this important clinical question. The blood sampling process from PDAC patients adhered to the protocols established by the clinical studies. It's important to clarify that the core objective of collecting blood samples from PDAC patients, which lies beyond the scope of this study, is centered around identifying a circulating biomarker capable of predicting the outcomes of FOLFIRINOX therapy. Our primary aim involves swiftly identifying a circulating biomarker during the treatment course, a task reflected in the timing of our blood sample collection points. To effectively achieve this overarching goal, we adopted a range of techniques to assess the blood samples. This study's central focus lies in shedding light on the influence of the chosen techniques on the resulting clinical conclusions.
In addition to this explanation, we hold the view that further accumulation of blood samples from the same patients would not yield more substantial or improved outcomes. The two collected samples effectively demonstrated that the chosen techniques possess the potential to impact the clinical conclusions. Therefore, the incorporation of additional samples would not significantly enhance the insights drawn from the study.
# Figures 2, 3 and 4: The flow cytometry and targeted gene expression data should be statistically analysed using correlation techniques.
We have opted for a broader scope in comparing the two techniques. Nevertheless, it's important to note that we have conducted numerous correlation tests, as illustrated in Supplementary Figure 6. These tests effectively convey the primary message of the manuscript. While we acknowledge the possibility of conducting correlation tests for all genes across all cells, it's worth considering that this would substantially inflate the number of tests performed and might not necessarily yield more insightful outcomes.
Furthermore, the immunological findings should also be correlated with the in-situ analysis of the tumour specimen.
We agree with the reviewer, the clinical picture is more complete when blood samples and tumor tissue samples are investigated using the same technique. Nevertheless, it's vital to consider that approximately 70% of PDAC patients fall into the category of being irresectable, rendering tumor tissue specimens inaccessible for analysis. Furthermore, acquiring pre- and post-cycle FOLFIRINOX therapy biopsies from PDAC patients isn't feasible due to the invasive nature of the procedure, which precludes authorization. These constraints underline the heightened significance of identifying a circulating biomarker capable of predicting therapeutic effects—a focal point we emphasized in our earlier response.
Discussion: With regard to the results, the authors should discuss the findings at the cellular and molecular level in more detail. Specifically, the increase in the immature system could be interpreted as a "normal" response to the cytotoxic effects of FOLFIRINOX chemotherapy combined with lipegfilgrast. On the other hand, the decrease at the mRNA level could be interpreted as an immunosuppressive function of the chemotherapy used.
We thank the reviewer for this comment, we have tried to elaborate more on the biological aspect, while keeping the technical note in mind, in our discussion.
Finally, the authors should mention that the results must be related to deep in-situ characterizations of tumour specimens.
As we wrote previously, we share the same opinion with the reviewer and think that correlating the blood immune profile to that of tissue samples is highly important. However, it’s not feasible for all PDAC patients and it’s out of the scope of this work.

Round 2
Reviewer 1 Report
Concerns remaining:
- The authors must present the results of the 18 populations measured by flow in frequency (%) among total leukocytes, and compare the results before and after treatment. The differences obtained before and after treatment in terms of cells per microliter, are due to the increase of leukocytes induced by lipegfilgrastim.
It is crucial to understand if the frequency of B cells and T cells decrease which will be on line with the decrease observed in gene expression.
Response from the authors:
We have decided to perform the suggested analysis as we agree that the total number of leukocytes might influence the conclusions drawn from the data. Nevertheless, especially regarding B and T cells, the large number of Granulocytes and Monocytes present in the samples skew the lower number of B and T cells when correcting for the total number of cells.
I didn´t see in the manuscript the results of the frequency of the 18 populations measured by flow.
- In legend of figure 3, the authors wrote that the abundance of mast cells (yellow) was significantly increased after treatment, but it looks on contrary. Moreover, the results on mast cells are not in agreement to those presented in supplementary figures, in terms of significance. In the same legend (D) the authors said that “the abundance of monocytes (green) including the subtypes was significantly increased after treatment”, but we can´t follow this information in the graphs, since we only have the information about total monocytes and also from conventional dendritic cells 2, and which means before and after in parenthesis at the top of the graphics.
We would like to thank the reviewer for pointing out the contradictions in our manuscript. We have double-checked the numbers and have adjusted the legend and Supplementary Figure 3 accordingly. Regarding Figure 3D the parenthesis refers to different monocyte definitions based on pairwise similarity in the before and after treatment group. This should now be more apparent by providing the cell definitions in Supplementary Table 2.
I am not able to find supplementary table 2
